# Galacto-Oligosaccharides Increase the Abundance of Beneficial Probiotic Bacteria and Improve Gut Architecture and Goblet Cell Expression in Poorly Performing Piglets, but Not Performance

**DOI:** 10.3390/ani13020230

**Published:** 2023-01-08

**Authors:** Adam Lee, Stephen C. Mansbridge, Lu Liang, Ian F. Connerton, Kenneth H. Mellits

**Affiliations:** 1Division of Microbiology, Brewing and Biotechnology, School of Biosciences, Sutton Bonington Campus, University of Nottingham, Loughborough, Leicestershire LE12 5RD, UK; 2Agriculture and Environment Department, Harper Adams University, Shropshire TF10 8NB, UK

**Keywords:** pigs, microbiota, suckling, galacto-oligosaccharides, probiotics, histology, goblet cells, milk replacer, lactic acid bacteria

## Abstract

**Simple Summary:**

New-born piglets often fail to thrive during suckling without any obvious signs of clinical disease, which causes significant economic loss and suffering. This is, in part, caused by enhanced production values, with too many piglets produced for the sow to provide adequate nutrition to its offspring. These piglets may not be receiving enough milk from the sow and may be removed to controlled environment pens and fed a complete commercial milk replacer to provide adequate nutrition and enhanced care. However, milk replacers do not traditionally contain the milk sugars found in sow milk that stimulate the development of a healthy immune system and gut microbiota. In this study, the effects of supplementing milk replacer with simple milk sugars on gut health, the microbiome and immune-protective goblet cells were investigated. Commercial milk replacer supplemented with milk sugars significantly increased the abundance of beneficial gut bacteria, improved gut health and the numbers of protective immune goblet cells. Results indicate that milk sugars given in milk replacer may be a useful addition in the husbandry of non-thriving, poorly performing piglets when moved to environmentally controlled pens away from sows and fit siblings by modulating their microbiome and gut health performance.

**Abstract:**

Poorly performing piglets receiving commercial milk replacers do not benefit from the naturally occurring probiotic galacto-oligosaccharides otherwise found in sow milk. Study objectives were to investigate the effects of complete milk replacer supplemented with galacto-oligosaccharides on the microbiome, gut architecture and immunomodulatory goblet cell expression of poorly performing piglets that could benefit from milk replacement feeding when separated from sows and housed with fit siblings in environmentally controlled pens. The study is novel in that it is one of the first to investigate the effects of supplementing complete milk replacer with galacto-oligosaccharides in poorly performing piglets. Gastrointestinal tract samples were collected from piglets, and the microbiome composition was assessed by 16s ribosomal ribonucleic acid gene sequencing. Gut architectural features, villus/crypt ratio and enumeration of goblet cells in tissues were assessed by histopathological techniques. The most abundant taxa identified at the genus level were *Lactobacillus*, *Streptococcus*, *Prevotella*, *Lactococcus* and *Leuconostoc*. Milk replacer plus galacto-oligosaccharides significantly improved gut architectural features and villus/crypt ratio throughout the gastrointestinal tract, increased the number of goblet cells and revealed a differential abundance of beneficial probiotic bacteria, particularly *Lactobacillus* and *Bifidobacterium*. In these respects, galacto-oligosaccharide-supplemented milk replacer may be a useful addition to animal husbandry in poorly performing, non-thriving animals when moved to environmentally controlled pens away from sows and fit siblings, thereby modulating the microbiome and gastrointestinal tract performance.

## 1. Introduction

The establishment and maintenance of beneficial gastrointestinal tract (GIT) microbiota during suckling is essential for the future performance, growth, health and welfare of animals [1]. Moreover, the GIT microbiota contributes to the developmental and metabolic needs of animals through short-chain fatty acid (SCFA) production, vitamin synthesis, complex carbohydrate digestion and immune system regulation [2,3]. However, poor pre-weaning performance and failure to thrive without obvious signs of clinical disease during lactation are of concern in piglet production, with estimated mortality rates of at least 12.6% representing significant economic loss [4], and this is notwithstanding gut dysbiosis due to known bacterial pathogens and/or scour from rotavirus [5]. Neonatal piglet viability has decreased in relation to selection for greater numbers of piglets born per sow. The focus on larger litter sizes has increased the number of piglets with decreased viability, lighter birth weights and a reduced ability to thrive in early life [6]. It is also recognised that weaning weight has a significant and profound effect on lifetime growth and performance, with lower birth-weight piglets achieving sub-optimal performance [7].

Microbiome manipulation through the addition of pre- and/or probiotic feeds, without pharmaceutical zinc oxide, particularly in healthy post-weaning pigs, is established, with an emphasis on beneficial lactic acid fermenting taxa, notably *Lactobacillus*, *Lactococcus*, *Leuconostoc*, *Bifidobacterium* and *Streptococcus* [8]. However, these approaches require further investigation in pre-weaning piglets with below-expected performance and economic return. Commercially available prebiotic galacto-oligosaccharides (GOS) comprising two to eight polymerised galactose units with terminal glucose moieties are functionally similar to those of mammalian milk and modulate gut architecture and intestinal microbiota in healthy pigs [9]. GOS is a major component of mammalian milk [10,11], which stimulates the development of the microbiota in neonates whilst conferring a variety of health benefits, including innate and adaptive immune development [12,13]. Naturally occurring milk GOS are typically composed of three to ten monosaccharide units, including galactose (Gal), glucose (Glc) and N-acetyl-glucosamine (GlcNAc), with some fucose and sialic acids. The core moiety present at the reducing end of milk oligosaccharides is either lactose (Gal(β1–4)Glc) or N-acetyl-lactosamine (Gal(β1–4)GlcNAc). Most animal milk oligosaccharides are sialylated (sialic acid at glycoprotein terminal ends), containing N-acetylneuraminic acid (Neu5Ac) and/or N-glycolylneuraminic acid (Neu5Gc) [14,15]. 

Porcine milk oligosaccharides (PMOs) contain the highest percentage of neutral oligosaccharides (20 %) in comparison with other domestic farm animals, the most abundant variety of mono and di-sialylated large oligosaccharides, and are most similar to the composition of human milk oligosaccharides [11]. However, PMOs decrease in abundance by approximately 43% during the first week of lactation, with the relative concentration of acidic PMOs decreasing and neutral PMOs increasing [16], suggesting a change in functionality during lactation and possibly the need for GOS early rather than later in life. 

In pigs, GOS is fermented by GIT bacteria, increasing beneficial probiotic populations and SCFA concentrations [9,17,18]. Studies have also shown that GOS inhibits pathogen adhesion and colonisation and reduces the expression of pro-inflammatory cytokines [9,18,19]. A healthy, well-differentiated intestinal mucosa has long, regular villi and high villus-to-crypt ratios [20], with the villus epithelium lining consisting mainly of absorptive enterocytes and specialised secretory goblet cells (GCs) [21]. GCs throughout the entirety of the GIT secrete mucins, forming a protective mucus layer against enteric pathogens. That is, they are critical to barrier function, maintenance of intestinal homeostasis, integrity of the GIT epithelium and physical lubrication of luminal contents [21,22]. GCs also form GC-associated antigen passages (GAPs) that deliver soluble luminal antigens to lamina propria dendritic cells [23]. GC intrinsic sensing of the GIT microbiota is now considered to play a critical role in regulating the exposure of the immune system to microbial challenges [24,25]. 

However, the modulatory effects of probiotics and prebiotics on GCs in suckling pigs remain largely uninvestigated. It is suggested that *Bacillus* probiotics can modulate and enhance GC function in *E. coli*-challenged pigs [26]. In vitro, GOS is considered to enhance mucosal barrier function through direct modulation of GC function and upregulation of GC secretory product genes [27,28]. In vivo, daily GOS dosing in piglets improves barrier function and relieves colonic inflammation via modulation of the mucosal microbiota [29]. Nevertheless, more in vivo studies are required to investigate the effects of GOS on the piglet microbiome, gut architecture and GC expression throughout the GIT, particularly in poorly performing pigs. 

For animals receiving sub-optimal nutrition from the sow, it is industry practice to remove them from farrowing pens and feed with a commercial milk replacer in controlled pen environments with no access to the sow. In contrast to natural sow colostrum and milk, commercial milk replacers have not traditionally contained GOS, although studies have demonstrated that formulas supplemented with GOS are safe and well tolerated in neonatal piglets [30]. Therefore, the objectives of this study were to investigate the effects of GOS on the microbiome, gut architecture and GC expression in poorly performing (non-thriving) piglets with below average birthweight who may benefit from milk replacement feeding alone or supplemented with GOS in four separate and repeated trials.

## 2. Materials and Methods

### 2.1. Ethics Approval Statement

All methods were carried out in accordance with the relevant guidelines and regulations of the Harper Adams University Research Ethics Committee and approved by them. The study is reported in accordance with ARRIVE guidelines. All animals were sacrificed using a schedule 1 method, the UK Animals (Scientific Procedures) Act 1986.

### 2.2. Animals and Trial Design

For all trials, piglets were derived from five to seven Landrace x Large white sows (JSR 9T; JSR Genetics, Driffield, UK) of similar parity (1–4), which were artificially inseminated with JSR 900 sire line semen (JSR Genetics, Driffield, UK). On day 100 (±3 d) of gestation, sows were moved to individual 1.81 × 2.61 m farrowing pens with a 2.2 × 0.63 m farrowing crate. Sows were housed in a single facility in separate pens. Trial 1 was conducted during March 2018, Trial 2 August 2018, Trial 3 October 2018 and Trial 4 March 2019 under identical housing and environmental conditions. Farrowing pens contained a 1.2 × 0.47 m piglet box heated with an industry standard heat lamp. Sows received a wheat-based lactation diet (BOCM Pauls Ltd., Wherstead, UK) containing 20.1% protein, 5.5 % oil, 3.5% crude fibre, 5.3% ash, 1.15% lysine, 3.1% methionine, 7.0% calcium, 1.6% phosphorous and 1.9% sodium plus water ad libitum. For prevention of iron deficiency, new-born piglets received a 1 mL intramuscular iron injection of 200 mg/mL (Ferroferon, Iron4u, Holte, Denmark) 24 h after birth. Sows were vaccinated with Porcilis^®^ Porcoli Diluvac Forte suspension for injection (Intervet International BV; Vm:EU/2/96/001/003-008) 3-weeks prior to farrowing for the passive immunisation of piglets by active immunisation of sows/gilts to reduce mortality and the clinical signs of neonatal enterotoxicosis. No vaccinations were given to the experimental animals directly. Piglets were ear-tagged on day 1 for identification. Poorly performing piglets potentially receiving sub-optimal nutrition from the sow were selected within the first seven days of life by visual assessment and the appearance of “non-thriving” by qualified animal technicians. This was based upon poor weight gain, a high degree of contamination with faecal material, the presence of watery faeces and overall health. Animals displayed no clinical symptoms of underlying disease, for example, scour or lameness, but were considered to benefit from a complete milk replacement feeding program. Piglets were group housed in 2.3 × 0.89 m slatted plastic isolation pens heated by industry standard lamps, with water ad libitum through a nipple drinker and twice daily feeding to appetite with either complete porcine milk replacer (CMR) (Faramate, Volac International Ltd., Royston, UK) alone (Appendix A) or supplemented with 1% (*w*/*w*) DP2 + GOS (Nutrabiotic^®^, Saputo Dairy UK, Weybridge, UK) with no access to a sow. Diets were designed to meet or exceed the nutrient requirements recommended for piglets. An acidifier (benzoic acid) was not included as a preservative in CMR due to possible interferences with the microbiome. Metal chain toys with plastic balls were provided in orphan pens as environmental enrichment. Piglets did not receive any creep feed supplementation, growth promoter or any other prophylactic antibiotic treatment during the studies. The temperature was kept within the range of 18–20 °C for sows and 23–24 °C for piglets, with light periods from 8:00 am to 16:30 pm. Piglets were weighed within 24 h of birth, on the day of recruitment to the trial (within 3–7 days of life) and then at weekly intervals terminating at the time of euthanasia when they were not considered suitable for economic production (week 4 of life). Each pen had a dedicated weight bucket to avoid microbiome contamination. Pens were deep cleaned by spraying with TopFoam, pressure washing and disinfection with MegaDes Novo (both from MS Schippers, Hapert, Netherlands) and left to dry. Pens were not used for any other experiments in between trials. Milk input and output, on a pen basis, were recorded daily and daily feed intake (DFI) was calculated.

### 2.3. Sample Collection and DNA Extraction

For trials 1 to 4, samples of digesta were aseptically collected post-mortem from piglets at anatomical sites throughout the GIT. Duodenal, jejunal, ileal, colonic, caecal and rectal lumen samples were held on dry ice prior to transfer to the laboratory and storage at −80 °C until bacterial DNA isolation. Bacterial DNA was isolated from 200 mg of luminal contents for each sample using the MP Biomedicals Fast DNA Kit for Feces (MP Biomedicals, Solon, OH, USA) according to the manufacturer’s instructions.

### 2.4. PCR Amplification of 16S rRNA Gene Sequences

Using the isolated DNA as a template, the V4 region of the bacterial 16S rRNA genes was PCR amplified using primers 515f (5′ GTGCCAGCMGCCGCGGTAA 3′) and 806r (5′ GGACTACHVGGGTWTCTAAT 3′) [31]. Amplicons were sequenced on the Illumina MiSeq platform (Illumina, Inc., San Diego, CA, USA) using 2 × 250 bp cycles according to the MiSeq Wet Lab SOP [32] separately for trials 1 to 4. Sequence data were deposited in the NCBI database within the Bioproject PRJNA866473, with SRA records available at: (https://www.ncbi.nlm.nih.gov/sra/PRJNA866473, accessed online 5 August 2022).

### 2.5. Microbiota Diversity Analysis

For each trial, the 16S rRNA gene sequence analysis was performed using mothur v.1.46.1, using default settings [33]. Analysis was performed according to the MiSeq SOP (https://www.mothur.org/wiki/MiSeq_SOP, accessed online 9 February 2022) [32]. The 16S rRNA gene sequences were aligned against a reference alignment based on the SILVA rRNA database [34] for use in mothur (release 132), available at: (https://www.mothur.org/wiki/Silva_reference files, accessed online 9 February 2022). The similarity cutoff for OTUs was 0.03. The consensus taxonomy of the OTUs was generated using the “classify.otu” command in mothur with reference data from the Ribosomal Database Project (version 14) [35,36] adapted for use in mothur available at: (https://www.mothur.org/wiki/RDP_reference_files, accessed online 9 February 2022). The relative abundance of OTUs annotated to taxa at the phylum and genus level were analysed from mothur output files using bespoke code written in R v4.1.1 using R Studio (2021.09.0) [37] and deposited at: (https://github.com/AdamLeeNottinghamUniversity/Piglets, accessed online 12 September 2022)

### 2.6. Histology

For all trials, immediately after excision, jejunal, ileal, colonic and caecal tissue samples from each piglet were fixed in 10% neutral buffered formalin in 40 mL prefilled specimen jars (Leica Microsystems UK, Ltd., Milton Keynes, UK). These were dehydrated through a series of alcohol solutions, cleared in xylene and embedded in paraffin wax. Sections 3 to 5 µm thick were prepared and stained with either haematoxylin and eosin (HE) to elucidate villus crypt architecture or periodic acid-Schiff (PAS) staining to enumerate mucin-producing GCs (VPG Histology, Bristol, UK). After staining, all HE and PAS slides were scanned using the NanoZoomer digital pathology system (Hamamatsu, Welwyn Garden City, UK). For the jejunum and ileum, measurements of villus height, crypt depth and villus and crypt area were made using the NanoZoomer digital pathology image programme (Hamamatsu). Ten well-oriented villi and crypts per tissue section of each piglet GIT sample from each trial were scanned at 40× resolution. Villus height was measured from the tip of the villus to the crypt opening, with the associated crypt depth measured from the base of the crypt to the level of the crypt opening. The villus/crypt ratio (VCR) was calculated by dividing the villus height by the crypt depth. The GCs were enumerated from ten well-oriented villi and crypts of jejunal, ileal, colonic and caecal sections stained with PAS, and the area of each was measured with individual GCs counted and pinned on each slide. For all tissue samples, both HE and PAS, well oriented villi and crypts were chosen using a random number generator at: (https://www.random.org/, accessed online 7 October 2019) from 1 to 10.

### 2.7. Statistical Analyses

Good’s coverage [38] and α-diversity expressed as Inverse Simpson diversity [39], and Chao richness [40] were calculated using the “summary.single” command in mothur [33]. Shapiro–Wilk tests [41] were used to determine normality for piglet weights at 24 h post-partum, day of trial 1, 7, 14 and 21, total weight gain, ADG and α-diversity metrics. Significant differences were tested using ANOVA with repeated measures for weight. Kruskal–Wallis tests were used to test for differences in α-diversity metrics. Estimates of β-diversity were calculated in mothur as Yue and Clayton dissimilarity (θ_YC_) [42], Bray–Curtis dissimilarity [43], and Jaccard similarity [44]. Analysis of molecular variance executed in mothur (AMOVA) was used to test for differences in β-diversity between samples [45,46]. Linear discriminant analysis effect size (LEfSe) was used to examine differential OTU abundances in mothur [47]. Where appropriate, multiple comparisons (ANOVA and AMOVA) were adjusted for false discovery rates using the Benjamini and Hochberg procedure [48]. All post-mothur statistical analyses were performed in R Studio (2021.09.0) unless otherwise stated [37]. For histological sections, ileal and villus height, crypt depth, VCR and number of GCs per mm^2^ tissues were analysed by Wilcoxon rank sum exact tests.

## 3. Results

### 3.1. Production Criteria: Weight, ADG and DFI

This study was performed in a commercial facility with pigs destined to go through the full production process. There were a limited number of farrowing pens (28) and only two isolation pens for poorly performing piglets. The number of poorly performing pre-weaning piglets differed between trials and could not be predicted in advance. Differences in “n” across trials one to four arose from those piglets that were visually assessed as poorly performing and subsequently randomly allocated to either receive milk replacer without GOS or milk replacer plus GOS in milk-replacer pens. Four non-GOS-fed piglets and four GOS-fed piglets were included in trial 1. Five non-GOS-fed piglets and four GOS-fed piglets were included in trial 2. Four non-GOS-fed piglets and four GOS-fed piglets were included in trial 3. Five non-GOS-fed piglets and five GOS-fed piglets were included in trial 4. In total, eighteen “poorly performing” non-GOS-fed piglets were included in analyses as opposed to seventeen “poorly performing” GOS-fed piglets across trials 1 to 4. A further eight pigs were physically removed during the study period (and excluded from analysis) after meeting the set humane endpoint threshold or requiring antibiotic treatment. Piglet weights at 24 h after birth; trial days 1, 7, 14 and 21; total weight gain and ADG were normally distributed according to Shapiro–Wilk tests (*p* > 0.05 in each case) for non-GOS and GOS-fed piglets alike in all four trials. These metrics were not significantly different across trials one to four or between non-GOS and GOS-fed piglets (*p* > 0.05 in each case, ANOVA, Table 1). Weight significantly increased with time for all trials and treatment groups (ANOVA, Day 1 to 21, *p* < 0.05 in each case). Piglet weights at 24 h *post-partum* significantly correlated with final weight at day 21 with no significant differences between treatment groups (linear modelling, *R*^2^ = 0.5, *p* < 0.001 for non-GOS piglets, *R*^2^ = 0.48, *p* < 0.001 for GOS-fed piglets, Appendix A). DFI was not significantly different between trials one to four for non-GOS-fed piglets and GOS-fed piglets alike (ANOVA, *p* = 0.709 and *p* = 0.343, respectively). The mean DFI (SD) for non-GOS-fed piglets increased from 0.311 (0.135) kg/piglet on day 1 of trials to 1.17 (0.476) kg/piglet on day 21 of trials. For GOS-fed piglets, the mean DFI (SD) at day one of trials was 0.493 (0.444), increasing to 1.547 (0.447) kg/piglet on day 21 of trials (Appendix A). Mean DFI and feed conversion ratio (FCR) across all four trials were not significantly different between treatments (*t*-test, *p* = 0.802 and *p* = 0.783, respectively).

### 3.2. GIT Microbiota Diversity

A total of 2,380,409 high-quality V4 16s rRNA sequence reads were obtained from 210 piglet GIT samples, with a Good’s coverage of 97.1 to 99.8% (minimum to maximum across all samples and four separate trials). The number of high-quality sequences obtained from each trial 1 to 4 and each section of the GIT, from the duodenum to the rectum, is shown in Appendix A. Inverse Simpson diversity and Chao richness were normally distributed for the majority of samples except for rectal samples from GOS-fed pigs in trial 1 (*p* = 0.042 and *p* = 0.006, respectively, Shapiro–Wilk tests); inverse Simpson diversity for ileal samples in non-GOS-fed piglets in trial 2 (*p* = 0.01); Chao richness for ileal samples from non-GOS-fed piglets in trial 3 (*p* = 0.04); inverse Simpson diversity for ileal samples from GOS-fed piglets in trial 4 (*p* = 0.01) and Chao richness for jejunal samples from GOS-fed piglets in trial 4 (*p* = 0.05). Inverse Simpson diversity and Chao richness (Table 2) were significantly different between trials one to four in some but not all comparisons (Kruskal–Wallis tests). Chao richness for duodenal samples from non-GOS piglets was significantly different across trials one to four (*p* = 0.015) as was the inverse.

Simpson diversity for GOS-fed piglets (*p* = 0.03); for jejunal samples from non-GOS piglets, Chao richness (*p* = 0.008); for colonic samples from GOS-fed piglets, inverse Simpson diversity (*p* = 0.02); for rectal samples, Chao richness for GOS-fed piglets (*p* = 0.03). Alpha diversity significantly increased from duodenal to rectal samples throughout the GIT in trials 1 to 4 (*p* < 0.05 in each case, Kruskal–Wallis tests) and in non-GOS piglets and GOS-fed piglets. There were exceptions for the trial 2 inverse Simpson diversity and Chao richness tests in non-GOS-fed pigs and Chao richness in GOS-fed piglets and the trial 4 Chao richness test for GOS-fed piglets. There were no significant differences in alpha diversity between non-GOS-fed piglets and GOS-fed piglets in trials 1 to 4. Table 3 shows the results of AMOVA used to test for significant differences in calculated β-diversity measures between trials one to four for non-GOS and GOS-fed piglets. All trials showed at least one significant difference in AMOVA for one or more β-diversity measures between non-GOS and GOS-fed piglets (*p* < 0.05 in each case). Complementary PCA plots for β-diversity are shown in Appendix A.

### 3.3. GIT Microbiota Composition

The number of unique, high-quality sequences clustered into OTUs obtained from each trial and each section of the GIT, from the duodenum to the rectum, is shown in Appendix A without discriminating between non-GOS-fed and GOS-fed piglets. Analyses were based on all sequences so that OTU numbers were consistent across all trials. In total, 3274 unique OTUs were identified from 210 piglet GIT samples across trials 1 to 4 and all GIT sections. 1279 OTUs were identified in trial 1, 1726 in trial 2, 1273 in trial 3 and 1216 in trial 4. 1062 OTUs were identified from the duodenum, 1000 from the jejunum, 1182 from the ileum, 1339 from the caecum, 1474 from the colon and 1503 from the rectum across trials 1 to 4. Figure 1 shows the number of OTUs shared by each trial and for each section of the piglet GIT. Only 176 OTUs were shared between trials 1 to 4 in the duodenum (16.6%), 118 in the jejunum (11.8%), 155 in the ileum (13.1%), 245 in the caecum (18.3%), 272 in the colon (18.5%) and 296 in the rectum (19.7%). However, these OTUs accounted for at least 97.9% of the total relative abundance of bacteria taxa at the genus level in each case. When analysed, on a trial-by-trial basis, predominant phyla in trial 1 across all samples were *Firmicutes* (77.08%), *Bacteroidetes* (16.07%), *Proteobacteria* (3.34%) and *Actinobacteria* (3.03%). Unclassified bacteria accounted for 0.25% of sequences. In trial 2, predominant phyla were *Firmicutes* (69.98%), *Proteobacteria* (13.59%), *Bacteroidetes* (9.69%), *Actinobacteria* (3.97%) and *Fusobacteria* (1.22%). Unclassified bacteria accounted for 0.73% of sequences. In trial 3, the predominant phyla were *Firmicutes* (77.15%), *Bacteroidetes* (14.67%), *Actinobacteria* (4.077%) and *Proteobacteria* (3.32%). Unclassified bacteria accounted for 0.44% of sequences. In trial 4, the predominant phyla were *Firmicutes* (78.70%), *Proteobacteria* (10.00%), *Bacteroidetes* (5.98%), *Actinobacteria* (2.69%) and *Deferribacteres* (1.34%). Unclassified bacteria accounted for 0.44% of sequences. At the genus level for trial 1, the most abundant taxa identified were *Lactobacillus* (22.94%), *Streptococcus* (19.85%), *Prevotella* (8.83%), *Leuconostoc* (7.72%), *Megasphaera* (3.11%), *Veillonella* (2.74%), *Bacteroides* (2.53**%),**
*Phascolarctobacterium* (2.51%), *Alloprevotella* (1.83%) and *Ruminococcaceae* unclassified (1.53%). For trial 2, the most abundant taxa identified at the genus level were *Veillonella* (14.84%), *Lactobacillus* (11.68%), *Pasteurellaceae* unclassified (11.32%), *Streptococcus* (9.27%), *Leuconostoc* (6.50%), *Lactococcus* (3.20%), *Bacteroides* (2.98%), *Prevotella* (2.53%), *Phascolarctobacterium* (2.37%) and *Megasphaera* (2.31%). For trial 3, the most abundant taxa identified at the genus level were *Lactobacillus* (27.25%), *Streptococcus* (13.55%), *Prevotella* (5.91%), *Phascolarctobacterium* (4.59%), *Blautia* (3.48%), *Veillonella* (3.38%), *Subdoligranulum* (3.25%), *Alloprevotella* (3.03%), *Ruminococcaceae* unclassified (2.95%) and *Collinsella* (2.67%). For trial 4, the most abundant taxa identified at the genus level were *Lactobacillus* (25.50%), *Streptococcus* (8.57%), *Veillonella* (7.70%), *Leuconostoc* (7.39%), *Lactococcus* (6.84%), *Enterobacteriaceae* unclassified (6.52%), *Megasphaera* (3.57%), *Blautia* (3.24%), *Prevotella* (2.43%) and *Phascolarctobacterium* (1.97%). Relative abundances of the top ten bacterial taxa at the phylum and genus levels for GIT samples from trials 1 to 4 for non-GOS and GOS-fed piglets are shown in Figure 2 and Figure 3, respectively. Significant differences in the differential abundance of OTUs ascribed to bacterial taxa at the genus level were calculated using LEfSe for each trial. Table 4 shows a summary of results for LEfSe for those OTUs ascribed to lactic acid fermenting bacteria throughout the GIT for non-GOS and GOS-fed piglets, trials 1 to 4, these being *Lactobacillus*, *Lactococcus*, *Leuconostoc*, *Bifidobacterium* and *Streptococcus* and the relative abundance of each OTU in parentheses, with a cut-off of 0.1% relative abundance. In total, across all GIT sections and trials 1 to 4, twenty-five linear discriminant features for all lactic acid fermenting bacteria occurred with non-GOS-fed piglets as opposed to forty-seven for GOS-fed piglets. In total, thirteen linear discriminant features for *Lactobacillus* occurred in non-GOS-fed piglets as opposed to twenty-eight for GOS-fed piglets. For *Leuconostoc*, this was two in non-GOS-fed piglets and three in GOS-fed piglets. For *Lactococcus*, this was one in non-GOS-fed piglets and two in GOS-fed piglets. For *Streptococcus*, this was eight in non-GOS-fed piglets and six in GOS-fed piglets. Only one linear discriminant feature ascribed to *Bifidobacterium* occurred in non-GOS-fed piglets, in contrast to eight in GOS-fed piglets. There was variation between trials. In trial 1, nine linear discriminant features ascribed to lactic acid fermenting bacteria were identified for non-GOS-fed piglets compared with eight for GOS-fed piglets. In trial 2, this was four for non-GOS-fed piglets and eighteen for GOS-fed piglets. In trial 3, three for non-GOS piglets and three for GOS-fed piglets. In trial 4, nine for non-GOS-fed piglets and eighteen for GOS-fed piglets. However, these results need to be interpreted in consideration of the relative abundance of each OTU. A full description of all differentially abundant OTUs and linear discriminant effect size (LEfSe) ascribed to all bacterial taxa at the genus level for all GIT sections and each trial one to four are shown in Appendix A, with a cut-off of either the top ten taxa or relative abundance by 0.1%, whichever occurred first, and showing unique OTUs for each trial.

### 3.4. Histology and Gut Architecture

Measurement of gut architecture parameters was determined using HE-stained slides and the enumeration of GCs was performed using PAS-stained slides, examples of which are shown in Figure 4. Upper intestinal villus height, crypt depth VCR and the number of GCs per mm^2^ GIT tissues were not normally distributed in most cases in trials 1 to 4 according to Shapiro–Wilk tests (*p* < 0.05) and therefore analysed non-parametrically on a trial-by-trial basis. Results for gut architecture are shown in Figure 5. The jejunal villus height was significantly greater for GOS-fed piglets in trials 1, 3 and 4 (*p* = 0.005, *p* = 6.4 × 10^−16^, *p* = 5.3 × 10^−8^, Wilcoxon rank sum exact tests). Jejunal villus height was significantly greater for non-GOS-fed piglets in trial 2 (*p* = 0.005). Ileal villus height was significantly greater for GOS-fed piglets in trials 3 and 4 (*p* = 61.6 × 10^−6^ and *p* = 1.1 × 10^−10^, Wilcoxon rank sum exact tests). There were no significant differences in jejunal or ileal crypt depth between treatments. Jejunal VCR was significantly greater for GOS-fed piglets in trials 1, 3 and 4 (*p* = 0.034, *p* = 5.1 × 10^−15^, *p* = 1.1 × 10^−8^, Wilcoxon rank sum exact tests). Jejunal VCR was significantly greater for non-GOS-fed piglets in trial 2 (*p* = 0.004). Ileal VCR was significantly greater for GOS-fed piglets in trials 3 and 4 (*p* = 0.0007 and *p* = 1.2 × 10^−9^, Wilcoxon rank sum exact tests). The number of GCs per mm^2^ GIT tissue per trial was determined from PAS-stained slides as shown in Figure 6. The number of GCs per mm^2^ in the jejunal villus was significantly greater in GOS-fed piglets in trials 1, 3 and 4 (*p* = 0.004, *p* = 1.8 × 10^−12^ and *p* = 1.1 × 10^−9^, respectively; in the jejunal crypt of GOS-fed piglets in trials 1 and 3 (*p* = 3.3 × 10^−9^ and *p* = 3.2 × 10^−12^, respectively); in the ileal villus in trials 2, 3 and 4 (*p* = 0.035, *p* = 1.8 × 10^−12^ and *p* = 1.2 × 10^−9^ respectively); in the ileal crypt in trials 3 and 4 (*p* = 0.0008 and *p* = 1.9 × 10^−5^, respectively); in the colonic crypt in trial 2 (*p* = 7.5 × 10^−6^); and in the caecal crypt in trial 3 (*p* = 0.0008, Wilcoxon rank sum exact tests). Regardless of inter-trial variation, it was possible to pool data and plot the area of the jejunal and ileal villi versus height and the area of the jejunal, ileal, colonic and caecal crypts versus depth (Appendix A). There was a significant correlation between the jejunal and ileal villus area and height and jejunal, ileal, colonic and caecal crypt area and depth (*p* < 0.001 in each case using linear modelling). For pooled data, jejunal villus height was significantly greater for GOS-fed piglets as opposed to non-GOS-fed piglets (*p* = 1.7 × 10^−11^) as was ileal villus height (*p* = 2.1 × 10^−8^), caecal crypt depth (*p* = 1.8 × 10^−4^) and colonic crypt depth (*p* = 0.008, Wilcoxon rank sum exact tests). There were no significant differences in jejunal and ileal crypt depth between non-GOS-fed and GOS-fed piglets. Similarly, there was a significant correlation between the jejunal and ileal villus areas and the jejunal, ileal, caecal and colonic crypt areas with GC density expressed as number of GCs per mm^2^ sectioned GIT tissues (*p* < 0.001 in each case, linear modelling, (Appendix A). Moreover, the number of GCs per mm^2^ GIT tissue was significantly higher in GOS-fed piglets than non-GOS-fed piglets for the jejunal villus (*p* = 7.4 × 10^−6^), the jejunal crypt (*p* = 1.4 × 10^−4^), the ileal villus (*p* = 2.8 × 10^−5^), the ileal crypt (*p* = 3.8 × 10^−7^), the colonic crypt (*p* = 5.3 × 10^−3^) and the caecal crypt (*p* = 0.003, Wilcoxon rank sum exact tests).

## 4. Discussion

This study is novel in that it is one of the first to investigate the effects of supplementing complete milk replacer with galacto-oligosaccharides in poorly performing piglets. The objectives were to investigate if industry standard milk replacer supplemented with GOS as opposed to milk replacer alone affected the microbiome, gut architecture, GC expression and performance of poorly performing, non-thriving piglets who were unlikely to receive adequate nutrition from the sow. Studies were conducted in a commercial facility with animals destined for the food chain. However, there were only a limited number of farrowing pens and a limited number of experimental isolation pens for poorly performing piglets. Therefore, trials had to be replicated. There were no significant differences in piglet weight at 24 h *post-partum*, trial days 1 to 21, total weight gain and ADG between the four repeated trials (Table 1). Neither was mean DFI (Appendix A) significantly different between trials and/or treatments, indicating that performance, food intake and animal husbandry were consistent between trials. However, supplementation of milk replacer with GOS had no significant effect on performance, with the main predictor of end-of-study weight (d21) being weight at 24 h *post-partum* (Appendix A). By way of comparison, it is reported that healthy GOS-fed piglets of the same age as this study showed no significant difference in body weight compared with the control groups but a significant increase in ADG [49]. However, weight at day 1 was, on average, 1.55 kg and at day 21~6 kg, with ADG at over 0.2 kg per day, values were much higher than in this study and for all trials and treatments (Table 1). Similarly, healthy GOS-fed piglets challenged with lipopolysaccharide endotoxin showed no significant difference in body weight compared with control groups but a significant increase in ADG [50]. Weight at day 1 was over 1.5 kg and at day 14 was over 4 kg, with an ADG of over 0.2 kg per day, with values again being higher than in this study. However, these studies administered a known dose of GOS at 1 g per kg body weight by manual oral infusion with continued access to the sow, as opposed to this study where animals were provided with milk replacer supplemented with GOS ad libitum with no access to the sow. In this respect, the addition of maternal GOS may have had a significant effect in contrast to the animals in this study that were removed from sows. Moreover, the animals in this study were initially underweight and poorly performing, which may explain why GOS had no effect on performance in terms of final body weight and ADG. Indeed, higher energy diet intake in the grower phase does not improve the performance of low-birth-weight pigs, which are less efficient than their heavier counterparts and are unable to show compensatory gains [51]. Studies have indicated that dietary supplementation with GOS post-weaning increases performance, and in this respect, it cannot be ruled out that GOS could positively affect performance in studies of longer duration than 21 days for poorly performing piglets [52]. 

There were no significant differences in α-diversity between non-GOS and GOS-fed piglets in trials 1 to 4 (Table 2). Previous studies have shown that intervention with GOS pre-weaning does not affect α-diversity, but post-weaning significantly increases the ACE and Chao1 indices of colonic mucosal communities in pigs [53]. Similarly, there were no significant differences in the Shannon, Simpson, Ace or Chao1 indices reported for colonic mucosal communities in lipopolysaccharide challenged piglets fed GOS compared with controls [29]. In this study, there were significant differences between trials in terms of α-diversity (*p* < 0.05 in each case), indicating that data could not be pooled but rather analysed on a trial-by-trial basis. The variation in data does not allow data to be pooled, in contrast to the study by Lee et al. [54], where trials were rigorously repeated in highly controlled conditions with a view to pooling data for the sake of performance-related measures. However, α-diversity expressed as inverse Simpson diversity or Chao richness significantly increased from the duodenum to the rectum in trials and for non-GOS and GOS-fed piglets alike, indicating the establishment of more diverse communities throughout the lower and upper GIT, consistent with previous work [55,56]. In Lee et al. (2022) [54], reported colonic inverse Simpson diversity ranged from (mean ± SD) 14.29 ± 2.20 to 20.20 ± 7.94, values that are broadly comparable with this study’s range of 12.65 ± 2.81 to 27.41 ± 8.09. For colonic Chao richness, this was 1241.08 ± 171.68 to 1794.48 ± 250.51 compared with 179.42 ± 43.35 to 315.40 ± 95.66 for the present study. Similarly, at suckling, Chao richness was determined as 1240.3 for faecal samples [57], and 1757 for colonic samples [58]. This large difference in species richness may be explained by the “poorly performing”, non-thriving nature of the piglets, since it is recognised that lower-weight, intrauterine growth-restricted piglets have lower microbial diversity [59]. Lower GIT diversity may be a result of poor performance or possibly a function of removing piglets from the sow to a controlled environment where they are not subjected to the microbiome of maternal sows and fit siblings.

In contrast to α-diversity, there were significant differences in β-diversity as measured by three metrics: Yue and Clayton dissimilarity (θ_YC_) [42], Bray–Curtis dissimilarity [43] and Jaccard similarity [44] between non-GOS and GOS-fed pigs in trials 1 to 4 (Table 3 and Appendix A). Differences in β-diversity were most prevalent for trials 2 and 4, and for Jaccard similarity, indicating that GOS possibly had more of an effect on microbial community membership, rather than community structure. Nevertheless, results demonstrate that early-life GOS intervention modulated GIT microbial composition, as in other studies [49,53].

The number of OTUs shared by all four trials (duodenum 177, jejunum 118, ileum 166, caecum 246, colon 272 and rectum 296) accounted for 97.9 to 99.76 % of the total relative abundance of all taxa at the genus level (Figure 1). Whilst the number of unique and/or partially shared OTUs may be much larger than those common between all four trials, they only account for 0.24 to 2.1 % of the total relative abundance of taxa across GIT samples. It is suggested there is a core microbiota in the GIT of healthy pigs, which can be a potential target for nutritional regulation and benefit the growth and GIT health of the animal [60,61]. If the definition of a core GIT microbiota is accepted as those being present in 90 % of samples [60], then the number of OTUs shared between trials 1 to 4 may be considered core to the suckling piglet microbiota. However, the core microbiota in pigs may be defined as those that are resident in the GIT throughout the lifetime of the animal as opposed to those that are “stage associated” and only occur at certain growth stages such as suckling [62]. At the phylum level, *Firmicutes**, Bacteroidetes*, *Proteobacteria* and *Actinobacteria* were considered core to the lifetime pig GIT microbiome [62], in keeping with trials 1 to 4 of this study, where these phyla occurred in all samples. In nursing pigs, the three most abundant core genera were *Prevotella*, *Lactobacillus* and *Oscillispira*, with *Blautia* identified as a nursery stage-associated genus [61]. In this study, *Prevotella*, *Lactobacillus* and *Streptococcus* occurred in all samples, confirming them as core microbiota but *not Oscillispira*. Perhaps it is useful to consider not only the “core” and “stage” microbiota but also the “peripheral” OTUs, which occur in low abundances but nevertheless contribute to the diversity of the whole microbiome.

Analysis of the taxa annotated to OTUs at the phylum level in the intestinal contents of piglets fed milk replacer alone or milk replacer with GOS in trials 1 to 4 enabled comparison of the relative abundance of taxa present at anatomical sites throughout the length of the GIT (Figure 2). *Bacteroidetes* were more prevalent in lower rather than upper GIT samples as opposed to *Firmicutes,* which were more prevalent in upper GIT samples, as reported by Crespo-Piazuelo et al., 2018 [55]. Predominant phyla in all trials were *Firmicutes*, *Bacteroidetes*, *Proteobacteria* and *Actinobacteria* in keeping with previous observations [54,62,63]. *Proteobacteria* are found in abundance during suckling and decline post-weaning [64]. They usually include commensal but opportunistic and potentially pathogenic organisms from the genera *Campylobacter*, *Escherichia*, *Salmonella* and *Helicobacter* [65]. Although occurring at very low relative abundances, their presence highlights the potential for the development of gut dysbiosis considering that low diversity bacterial ecosystems have reduced colonisation resistance to pathogens [66]. The relative abundance of bacterial taxa annotated to OTUs at the genus level show *Lactobacillus, Streptococcus*, *Prevotella* and *Leuconostoc* were highly prevalent throughout all the trials (Figure 3). Although occurring throughout the GIT, *Streptococcus* was more prevalent in the upper GIT samples. LEfSe gives a clearer distinction between non-GOS and GOS-fed piglets (Table 4) and highlights those lactic acid fermenting and beneficial OTUs occurring between treatments and across the GIT for trials 1 to 4. Dietary supplementation with GOS increases *Lactobacillus* and *Bifidobacterium* populations in pigs, as confirmed in this study [9,52]. In trial 1, GOS significantly increased *Streptococcus* in the upper GIT as opposed *to Lactobacillus*, suggesting competition between these organisms. Nevertheless, GOS significantly increased *Bifidobacterium and Lactobacillus* in the lower GIT. Trials 2 and 4 identified a number of linear discriminant features attributable to all five lactic acid bacteria, confirming the lactogenic and bifidogenic effects of GOS throughout the GIT of suckling pigs.

Despite a large degree of variation between trials, this study has demonstrated that GOS significantly affects gut architecture and VCR in poorly performing piglets (Figure 5 and Appendix A). Not only does GOS protect against mucosal GIT damage in lipopolysaccharide-challenged pigs [29], but it also promotes higher villus height and VCR in *E. coli*-challenged pigs [67], suggesting physiological and nutritional health benefits for non-healthy animals and those with compromised upper GIT mucosal surfaces. However, the effects of GOS on production remain to be seen, presumably due to the non-thriving nature of animals in this study, the study length of 21 days and the inability of low-birth-weight animals to make compensatory weight gains [51]. Nevertheless, GC density per mm^2^ tissue significantly increased throughout the GIT in GOS-fed piglets compared with non-GOS-fed piglets (Figure 6 and Appendix A). This is an important finding since GCs are known to be essential to barrier function and immune regulation in animals [21,22,25]. The physiological effect of GOS on GIT architecture and GC expression is not limited to pigs, with significant increases in villus height, caecal crypt depth and increased GIT GC numbers found in GOS-fed chickens [68,69]. GOS directly modulates the expression of GC secretory products that contribute to the production of barrier-enhancing mucins via cell surface receptors [27]. However, this may be indirectly modulated by the microbiota, particularly the lactic acid-producing bacteria *Bifidobacterium* and *Lactobacillus* spp., as significantly increased in this study by GOS, which support intestinal cell regeneration and the proliferation of intestinal stem, Paneth and GCs through the action of lactic acid [70]. 

## 5. Conclusions

GOS, as a supplement to milk replacer formula for piglets separated from sows, is palatable and well tolerated, with significant increases in weight during trials, but no significant performance difference between treatments. Administration of GOS had no significant effect on α-diversity, for which Chao richness appeared to be low, but may be a function of inherent poor performance and/or removing piglets from sows and healthy siblings to controlled environments, thus possibly being a product of methodology. Nevertheless, GOS significantly modulated GIT microbial communities as demonstrated by β-diversity measures, with key effects on microbial community membership rather than structure, demonstrating that GOS is effective in promoting more diverse, beneficial communities. GOS significantly increased linear discriminant features attributable to lactic acid-producing bacteria, notably *Lactobacillus* and *Bifidobacterium* throughout the GIT, demonstrating benefits as a supplement to milk replacer in poorly performing piglets. GOS significantly improved GIT architectural features and VCR throughout the upper GIT as well as increasing the number of barrier-enhancing and immunomodulatory GCs, possibly through direct modulation by GOS or indirectly by the lactogenic effect of lactic acid-producing bacteria. The significance of these indicators lies in the improvement of GIT health in poorly performing animals, giving them a better chance of survival in controlled environments. In conclusion, GOS significantly increases the differential abundance of beneficial probiotic bacteria, particularly *Lactobacillus* and *Bifidobacterium*, and improves gut architecture and goblet cell expression in poorly performing piglets. In these respects, a GOS-supplemented milk replacer may be a useful addition to animal husbandry for poorly performing, non-thriving animals when moved to environmentally controlled pens away from sows and their thriving siblings, thereby modulating the microbiome and GIT performance. Future applications may include the addition of GOS in milk replacers for healthy piglets requiring additional nutrition from the sow, but this would require further research.

## Figures and Tables

**Figure 1 animals-13-00230-f001:**
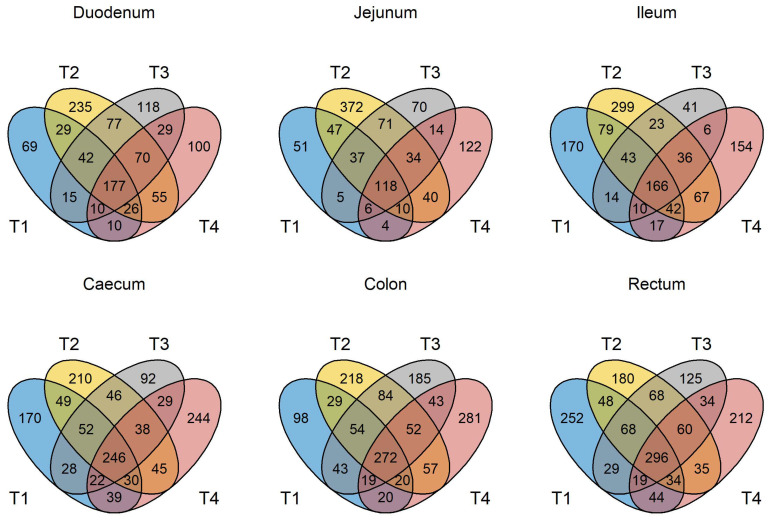
Venn diagram depicting unique and shared OTUs at the genus level in GIT samples from pigs in trials 1 to 4. T1 = trial 1; T2 = trial 2; T3 = trial 3; T4 = trial 4.

**Figure 2 animals-13-00230-f002:**
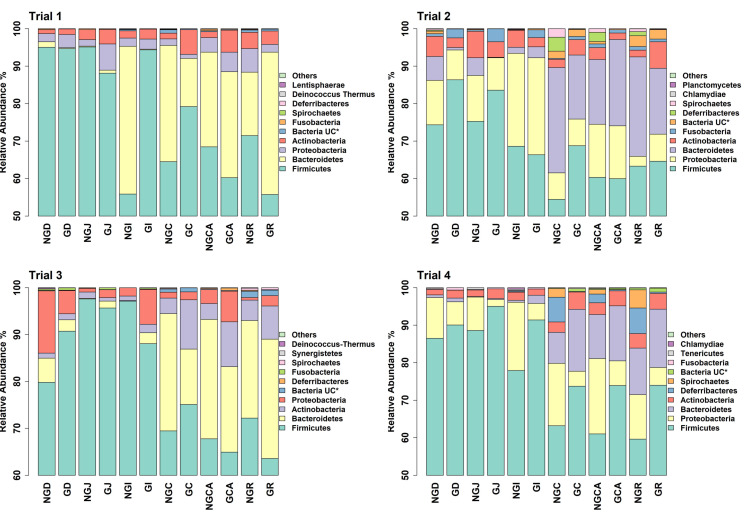
Relative abundance of bacterial taxa annotated to OTUs at the phylum level as identified from GIT samples of piglets fed milk replacer alone or milk replacer with GOS in trials 1 to 4. * UC = unclassified at the phylum level. NGD = Non-GOS Duodenum; GD = GOS Duodenum; NGJ = Non-GOS Jejunum; GJ = GOS Jejunum; NGI = Non-GOS Ileum; GI = GOS Ileum; NGC = Non-GOS Colon; GC = GOS Colon; NGCA = Non-GOS Caecum; GCA = GOS Caecum; NGR = Non-GOS Rectum; GR = GOS Rectum.

**Figure 3 animals-13-00230-f003:**
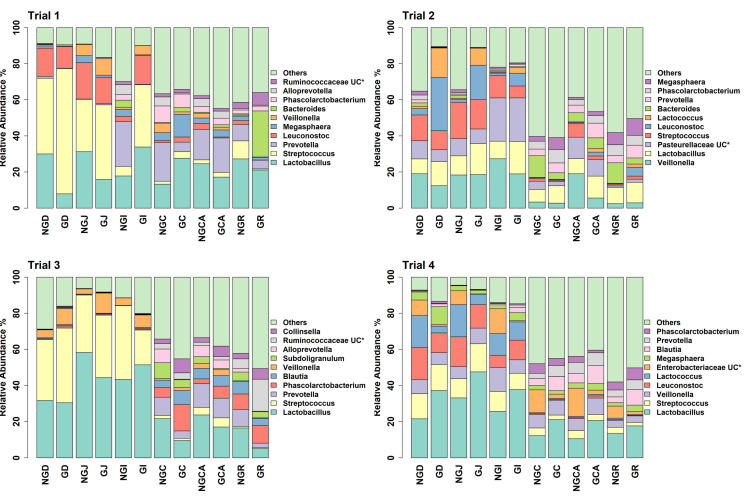
Relative abundance of bacterial taxa annotated to OTUs at the genus level as identified from GIT samples of piglets fed milk replacer alone or milk replacer with GOS in trials 1 to 4. * UC = unclassified at the genus level. NGD = Non-GOS Duodenum; GD = GOS Duodenum; NGJ = Non-GOS Jejunum; GJ = GOS Jejunum; NGI = Non-GOS Ileum; GI = GOS Ileum; NGC = Non-GOS Colon; GC = GOS Colon; NGCA = Non-GOS Caecum; GCA = GOS Caecum; NGR = Non-GOS Rectum; GR = GOS Rectum.

**Figure 4 animals-13-00230-f004:**
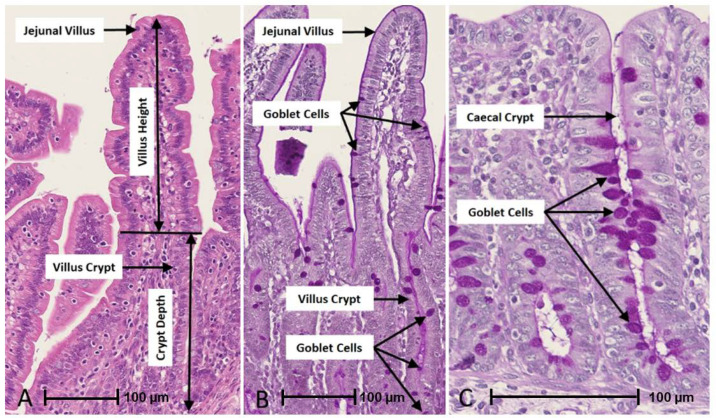
Example of gut architecture and GC enumeration in GIT sections taken with the NanoZoomer digital pathology system. (**A**) = HE-stained jejunal villus and crypt used for determination of villus height, crypt depth and VCR. (**B**) = PAS-stained jejunal villus and crypt used for GC enumeration. (**C**) = PAS-stained caecal crypt used for GC enumeration.

**Figure 5 animals-13-00230-f005:**
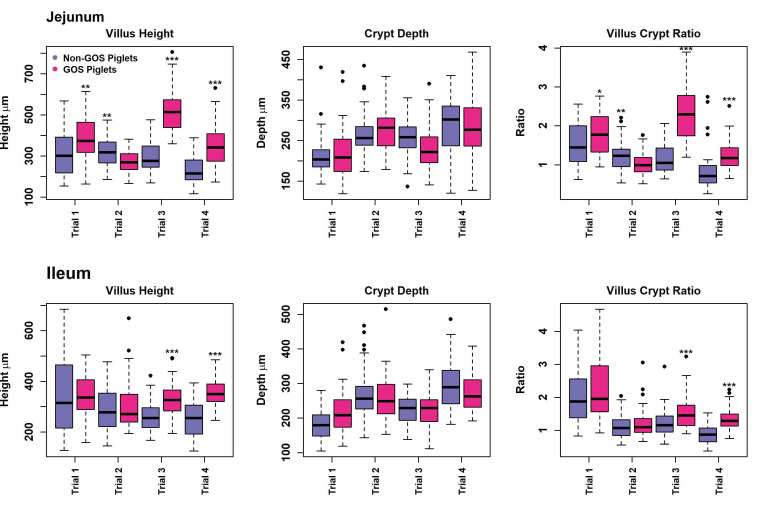
Upper intestinal gut architecture of non-GOS and GOS-fed piglets for trials 1 to 4 showing villus height, crypt depth and villus crypt ratio. (* *p* < 0.05, ** *p* < 0.01, *** *p* < 0.001, Wilcoxon rank sum exact tests). Dots show outliers above maximum interquartile range.

**Figure 6 animals-13-00230-f006:**
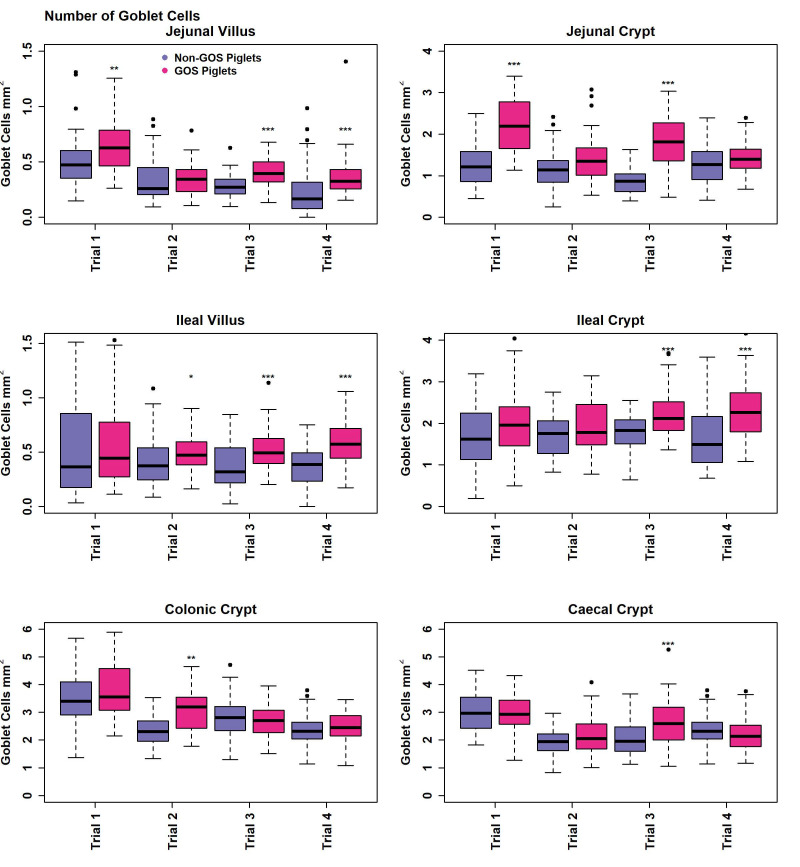
Number of goblet cells per mm^2^ GIT tissue of non-GOS and GOS-fed piglets for trials 1 to 4. (* *p* < 0.05, ** *p* < 0.01, *** *p* < 0.001, Wilcoxon rank sum exact tests). Dots show outliers above maximum interquartile range.

**Table 1 animals-13-00230-t001:** Pig weights at 24 h *post-partum*, days 1 to 21 of trials, total weight gain, ADG and FCR.

	Trial 1		Trial 2		Trial 3		Trial 4	
Weight kg	Non-GOS	GOS	Non-GOS	GOS	Non-GOS	GOS	Non-GOS	GOS
24 h	1.34 (0.19)	1.30 (0.20)	1.09 (0.37)	1.11 (0.28)	1.16 (0.25)	1.21 (0.24)	0.97 (0.15)	0.90 (0.19)
Day 1	1.48 (0.23)	1.43 (0.27)	1.15 (0.37)	1.22 (0.28)	1.41 (0.45)	1.36 (0.38)	1.12 (0.18)	1.16 (0.11)
Day 7	2.17 (0.43)	2.32 (0.26)	2.17 (0.72)	2.17 (0.52)	2.31 (0.66)	2.10 (0.48)	1.77 (0.37)	1.92 (0.53)
Day 14	2.93 (0.45)	3.23 (0.24)	3.44 (1.33)	3.26 (0.94)	3.25 (0.52)	2.74 (0.54)	2.35 (0.76)	3.13 (0.81)
Day 21	4.41 (0.85)	4.53 (0.20)	4.30 (1.73)	3.91 (1.44)	4.58 (0.44)	3.71 (0.75)	3.11 (0.99)	3.93 (0.92)
Total gain	2.93 (0.81)	3.10 (0.28)	3.15 (1.49)	2.69 (1.18)	3.17 (0.43)	2.35 (0.50)	1.99 (0.95)	2.77 (0.85)
ADG kg/d	0.15 (0.04)	0.15 (0.01)	0.14 0.07)	0.12 (0.05)	0.16 (0.02)	0.12 (0.02)	0.10 (0.04)	0.14 (0.04)
DFI kg	0.84 (0.40)	1.03 (0.44)	1.04 (0.38)	0.98 (0.30)	1.23 (0.41)	1.00 (0.25)	0.83 (0.17)	0.91 (0.29)
FCR	5.60	6.87	7.43	8.17	7.69	8.33	8.30	6.50

Values are means (SD). The means are not significantly different across trials or between non-GOS and GOS-fed piglets (ANOVA, *p* > 0.05, in each case). The mean weight significantly increased with time for all trials and treatment groups (ANOVA, Day 1 to 21, *p* < 0.01 in each case).

**Table 2 animals-13-00230-t002:** Alpha diversity. Mean (SD).

	Trial 1		Trial 2		Trial 3		Trial 4	
	Non-GOS ^3,4^	GOS ^3,4^	Non-GOS	GOS ^3^	Non-GOS ^3,4^	GOS ^3,4^	Non-GOS ^3,4^	GOS ^3^
Inverse Simpson Diversity								
Duodenum ^1,2^	5.11 (2.31)	2.93 (1.27)	21.16 (16.21)	6.52 (0.50)	6.28 (1.36)	5.13 (1.79)	7.47 (2.28)	14.08 (5.83)
Jejunum	8.12 (2.85)	6.36 (4.50)	17.39 (10.20)	7.60 (2.13)	5.76 (1.13)	4.99 (1.45)	8.49 (2.73)	9.61 (0.80)
Ileum	15.42 (10.12)	6.17 (3.18)	12.04 (9.18)	14.51 (10.88)	5.04 (2.28)	8.48 (3.67)	10.52 (3.53)	15.22 (7.29)
Caecum ^2^	17.56 (6.04)	20.63 (4.65)	29.31 (15.82)	22.64 (4.81)	18.96 (4.24)	17.22 (3.55)	21.49 (13.62)	20.44 (3.31)
Colon ^2^	18.89 (4.41)	12.65 (2.81)	38.27 (12.78)	27.41 (8.09)	21.56 (2.70)	14.03 (8.11)	21.36 (10.81)	25.32 (3.38)
Rectum	21.00 (9.77)	22.08 (12.31)	26.14 (7.96)	25.14 (5.94)	23.11 (5.76)	17.86 (3.53)	23.91 (11.07)	26.05 (1.46)
Chao Richness								
Duodenum ^1^	113.37 (43.90)	95.53 (11.59)	301.92 (99.28)	213.78 (43.28)	192.39 (61.81)	161.19 (64.40)	132.45 (25.48)	178.14 (59.61)
Jejunum ^1^	87.97 (10.65)	111.59 (34.05)	278.74 (76.24)	169.04 (38.75)	122.98 (37.74)	122.26 (63.35)	110.23 (8.81)	123.39 (36.81)
Ileum	207.53 (96.48)	104.44 (24.26)	256.66 (79.15)	248.67 (183.61)	116.42 (25.74)	146.11 (87.06)	184.46 (82.81)	177.95 (45.65)
Caecum	237.49 (70.67)	213.63 (32.34)	257.02 (64.16)	226.92 (62.83)	279.81 (52.19)	237.44 (23.60)	260.98 (66.12)	220.23 (55.32)
Colon	197.52 (49.21)	179.42 (43.35)	310.12 (49.66)	259.70 (60.27)	295.06 (49.14)	258.41 (20.57)	315.40 (95.66)	243.22 (47.65)
Rectum ^2^	264.23 (132.91)	261.21 (29.00)	324.84 (30.92)	286.94 (76.27)	307.68 (71.12)	314.12 (46.32)	243.34 (63.14)	156.87 (34.07)

^1^ Significant differences between trials 1 to 4 for non-GOS piglets (*p* < 0.05 in each case, Kruskal–Wallis tests). ^2^ Significant differences between trials 1 to 4 for GOS piglets (*p* < 0.05 in each case, Kruskal–Wallis tests). ^3^ Significant differences across GIT for inverse Simpson diversity (*p* < 0.05 in each case, Kruskal–Wallis tests). ^4^ Significant differences across GIT for Chao richness (*p* < 0.05 in each case, Kruskal–Wallis tests).

**Table 3 animals-13-00230-t003:** β-diversity showing significant differences in AMOVA between non-GOS and GOS-fed piglets in four separate trials.

Trial	Duodenum	Jejunum	Ileum	Caecum	Colon	Rectum
Yue & Clayton Dissimilarity (θ_YC_)						
1	0.246	0.293	0.373	0.517	0.173	0.031
2	0.012	0.015	0.284	0.011	0.047	0.313
3	0.486	0.513	0.085	0.494	0.392	0.034
4	0.062	0.209	0.185	0.005	0.05	0.121
Bray-Curtis Dissimilarity						
1	0.186	0.217	0.173	0.427	0.254	0.062
2	0.01	0.012	0.155	0.006	0.087	0.197
3	0.456	0.327	0.069	0.131	0.241	0.034
4	0.001	0.252	0.257	0.004	0.01	0.061
Jaccard Similarity						
1	0.062	0.660	0.09	0.106	0.106	0.241
2	0.031	0.025	0.126	0.016	0.140	0.006
3	0.456	0.017	0.145	0.034	0.034	0.034
4	0.018	0.006	0.173	0.005	0.005	0.012

**Table 4 animals-13-00230-t004:** Significant differences in differential abundance of lactic acid fermenting bacteria throughout the GIT for non-GOS and GOS-fed piglets, trials 1 to 4.

	Trial 1		Trial 2		Trial 3		Trial 4	
	Non-GOS	GOS	Non-GOS	GOS	Non-GOS	GOS	Non-GOS	GOS
Duodenum	Otu008 LB (2.71) * Otu015 LB (0.88) * Otu017 LB (0.70) * Otu024 LB (0.30) *	Otu005 SC (8.4) *	Otu024 LB (1.1) *	Otu002 LN (29.5) * Otu004 LC (16.3) * Otu017 LB (2.6) ** Otu038 BB (0.8) ** Otu074 LB (0.22) **	NS	Otu074 SC (0.11) *	Otu002 LC (15.9) * Otu003 SC (10.6) **	Otu018 LB (1.9) * Otu025 LB (1.1) ** Otu067 LB (0.11) *
Jejunum	Otu003 LB (14.6)* Otu005 LB (8.1) * Otu017 LB (1.4) *	Otu031 SC (0.26) *	NS	Otu003 LN (18.2) * Otu004 LC (9.4) * Otu023 BB (1.5) ** Otu024 LB (1.6) **	Otu017 LB (0.9) *	NS	Otu024 BB (0.56) ** Otu025 LB (0.48) * Otu028 LB (0.39) * Otu042 SC (0.21) **	Otu032 SC (0.36) * Otu035 LB (0.36) ** Otu036 LB (0.36) ** Otu054 LB (0.12) **
Ileum	NS	Otu011 SC (3.3) * Otu091 SC (0.13) *	Otu060 SC (0.15) **	Otu007 LB (5.1) ** Otu008 LB (4.9) ** Otu016 LB (2.3) * Otu030 BB (0.92) * Otu054 LB 0.30) **	Otu038 SC (0.17) *	Otu027 BB (0.3) *	Otu018 SC (2.5) * Otu045 SC (0.33) ** Otu062 LB (0.18) *	Otu005 LB (10.0) * Otu068 LB (0.20) **
Caecum	Otu016 LN (3.3) *	Otu020 BB (2.6) *	Otu058 SC (0.62) ** Otu135 SC (0.10) *	Otu006 LB (4.39) ** Otu134 BB (0.12) *	Otu117 LB (0.13) *	Otu031 LN (1.2) **	NS	Otu002 LB (9.7) * Otu003 LB (7.2) ** Otu008 LB (4.1) * Otu019 LB (2.25) * Otu020 LB (2.48) * Otu059 LB (0.48) * Otu076 LB (0.36) **
Colon	NS	Otu009 BB (4.1) ** Otu033 LB (1.1) *	NS	Otu003 LB (4.7) * Otu038 BB (0.54) *	NS	NS	NS	Otu003 LB (6.2) ** Otu020 LB (2.1) *
Rectum	Otu021 LN (1.7) *	Otu022 LB (1.6) *	NS	NS	NS	NS	NS	NS

LB = *Lactobacillus*; LC = *Lactococcus*; LN = *Leuconostoc*; BB = *Bifidobacterium*; SC = *Streptococcus*. * *p* < 0.05, ** *p* < 0.01, NS = no significant difference in differential abundance between non-GOS and GOS-fed piglets (LEfSe). Figures in brackets are the relative abundance of named Otu for each GIT section and trial.

## Data Availability

Sequence data were deposited in the NCBI database within the Bioproject PRJNA866473 with SRA records available at: (https://www.ncbi.nlm.nih.gov/sra/PRJNA866473, accessed online 5 August 2022). R-code for analyses are deposited at: (https://github.com/AdamLeeNottinghamUniversity/Piglets, accessed online 12 September 2022).

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
