# Peer review of "Galacto-Oligosaccharides Increase the Abundance of Beneficial Probiotic Bacteria and Improve Gut Architecture and Goblet Cell Expression in Poorly Performing Piglets, but Not Performance"

_animals, 2023, doi:10.3390/ani13020230_

Round 1

Reviewer 1 Report

This study investigated the effects of galacto-oligasaccharides on the growth performance, gut microbiota composition and gut health status of poorly performing piglets. It should be noted that the application of galacto-oligasaccharides experts no beneficial effects on growth performance of piglets, so what's the significance of the measurements for other indicators? And what's the applications of this product in the future?

Author Response

Reviewer 1

Firstly, we would like to thank the Reviewer for their comments. We take note that galacto-oligosaccharides exerted no beneficial effects on growth performance of piglets. We have already stated in the discussion (lines 457-459) “Moreover, animals in this study were initially underweight and poorly performing, which may explain why GOS had no effect on performance in terms of final body weight and ADG.” Further to this we have added (lines 459-461) “Indeed, higher energy diet intake in the grower phase does not improve the performance of low birth weight pigs, which are less efficient than their heavier counterparts and are unable to show compensatory gains [51].” We have added the reference of Douglas, Edwards & Kyriazakis, 2014. We have also stated (line 551-554) “However, the effects of GOS on production remain to be seen, presumably due to the non-thriving nature of animals in this study, the study length of 21 days and the in-ability of low birth weight animals to make compensatory weight gains [51].” We have added a sentence to state the significance of results (lines 584-586) “The significance of these indicators lies in the improvement of GIT health in poorly performing animals giving them a better chance of survival in controlled environments.” Furthermore, we have introduced the importance (significance) of the microbiota in lines 50 to 54 and 64 to 67. In terms of histology we have introduced the importance (significance) of villus crypt ratio and goblet cells lines 91-100. With regards to applications we have stated (lines 588-522) “…GOS supplemented milk replacer may be a useful addition to animal husbandry in poorly performing, non-thriving animals….” The application being supplementation of commercial milk replacer with GOS. We have also added (lines 592-593) “Future applications may include inclusion of GOS in milk replacers for healthy piglets requiring additional nutrition to the sow, but this would require further research.”

Reviewer 2 Report

Dear Editor,

The overall scientific soundness of the manuscript is very good. The manuscript may be considered for publication in its current form. The authors should, in my opinion, include the average DFI values of the piglets and the corresponding feed efficiencies in Table 1: (Pig weights at 24 hours post-partum, days 1 to 21 of trials, total weight gain and ADG.) L268 - 269.

Thank you.

Author Response

Reviewer 2

We thank the reviewer for their comments. We have, as suggested, included the average DFI values and FCR of the piglets in Table 1.

Reviewer 3 Report

The present manuscript focuses on the Galacto-oligosaccharides increase abundance of beneficial probiotic bacteria, improve gut architecture and goblet cell expression in poorly performing piglets.  The subject frame of the work is well constructed. So, in this respect and this article should be contributed to present research. I recommended this work for publication after the following minor revisions.

1.      I did not see any improvement in bioactivity, so I suggest to revised the tittle accordingly.

2.      There are several typographical mistakes as well in whole manuscript. Therefore, the author’s thoroughly careful check the language and typo mistake to minimize the error.

3.      The abstract should be beginning with a sentence about the background of concept and the aims as well as novelty of study should be mentions. What exactly is the novelty of this study? The abstract is poorly written and should be improved. Abbreviations must be avoided in abstract. Parenthesis should be avoided in abstract - this is poor writing. Please improve.

4.      All figures are of poor technical quality and not suitable for publication, especially in a high reputed journal. Font size and kind is too small and must be unified in all figures. Small writings are unreadable. All figures must be self-explanatory. Axis titles are poorly presented or absent. Units are missing. Are the data presented in figures significantly different? At least error bars should be shown.

5.      What is exactly the novelty of this review article, as so many articles were already out, is this the updates version or some other novel aspect. Author needs to revised it carefully and should provide novelty statement. 

Author Response

Reviewer 3

We thank the reviewer for their comments.

Point 1 “I did not see any improvement in bioactivity, so I suggest to revise the title accordingly.” Is the reviewer using the term “bioactivity” with regard to animal performance as measured by weight gain and average daily gain or in the wider sense of “any effect on, interaction with, or response from living tissue?” If this point is with reference to “performance”, then we are happy to modify the title to “Galacto-oligosaccharides increase abundance of beneficial probiotic bacteria, improve gut architecture and goblet cell expression in poorly performing piglets but not performance.”  We would be happy for the Editor to make a final choice of title, given that “performance” was not a primary indicator. We have made it clear in the manuscript why improvements in animal performance could not be expected given the short duration of trials, the poor performance of piglets and their inability to make compensatory weight gains. In this respect it may be useful for reviewer 3 to see our comments and corrections in relation to reviewer 1.

Point 2. We have spell checked the document and corrected any mistakes. We used British English spelling as opposed to American English spelling. We will ask the journal for their preference and amend if required.

Point 3. We have added sentences to describe concept, objectives and novelty as suggested by the reviewer. “Poorly performing piglets receiving commercial milk replacers do not benefit from the naturally occurring probiotic galacto-oligosaccharides otherwise found in sow milk. Study objectives were to investigate the effects of complete milk replacer supplemented with galacto-oligosaccharides on the microbiome, gut architecture and immunomodulatory goblet cell expression of poorly performing piglets that could benefit from milk replacement feeding when separated from sows and fit siblings in environmentally controlled pens. The study is novel in that it is one of the first to investigate the effects of supplementing complete milk replacer with galacto-oligosaccharides in poorly performing piglets.” We have removed abbreviations and parentheses from the abstract. Please note, there is a word limitation in the abstract and inclusion of the reviewer’s suggestions comes at the expense of other text.

Point 4. We have produced high resolution Tiff files for publication of figures which will be submitted to the journal when requested by them. The figures provided to reviewers by the journal were of lower quality and for review purposes only, not publication standard. The guidelines for Authors clearly states that figures and tables should be inserted in the body of the text as close as possible to their first reference in the text. We have attempted to do this, but the graphic quality for review is less than that finally submitted for publication. Figure captions are provided at the end of the manuscript according to journal instructions (lines 814-832). It is for the journal to “marry” captions with figures provided, which is why they are not embodied in figures but as an addendum to the manuscript which the reviewers see and for the journal to publish as they see fit, according to their formats. Figures are self explanatory once the figure captions are read. Figure captions, where appropriate, include P values showing levels of significance. Figures 1 to 6 which are in the main body of the text have the same font. We have increased font size for Figures 2, 3, 5 and 6. We do not agree that axes titles are absent in the main figures. Given the non-parametric nature of data, particularly microbiome and histology data, it would not be appropriate to use error bars. For supplementary figures S3A to S3D we have increased the font size. We think the reviewer wishes to have the individual graph headings (Duodenum, Jejunum, Ileum, Caecum, Colon, Rectum) on every PCoA plot. We can do this, but it does mage the graphic look crowded. We would appreciate an editorial decision on this. We have increased font size for figures S5 and S6.  

Point 5. The reviewer states “What is exactly the novelty of this review article, as so many articles were already out, is this the updates version or some other novel aspect. Author needs to revise it carefully and should provide novelty statement.” We would like to clarify that the paper is not a review article, but an article of genuine research that is not an update of any previous research we have carried out, or that based upon previous studies by other researchers. Our concept was novel, in regards to feeding poorly performing piglets with milk replacer supplemented with galacto-oligosaccharides. We agree, there are plenty of studies on the effects of galacto-oligosaccharides in animals. However, the study designs and objectives are generally considerably different than our own. The novelty of this study lies in two major factors. 1) Supplementing complete milk replacer with galacto-oligosaccharides. 2) Feeding this to non-thriving poorly performing piglets. To our knowledge, this would be one of the first studies to investigate the effects of supplementing complete milk replacer with galacto-oligosaccharides in poorly performing piglets. We have added a “novelty statement” to the paper as suggested by the reviewer (lines 429-430) “This study is novel in that it is one of the first to investigate the effects of supplementing complete milk replacer with galacto-oligosaccharides in poorly performing piglets.” We would be happy to reference commercial brands that are used as “complete milk replacers” for those animals not receiving adequate nutrition from the sow (if this allowed by the journal). Whilst efficacious, these formulae do not contain galacto-oligosaccharides, naturally present in sow milk which are known to promote immune, microbiome and histological benefits in healthy animals. Our point is, that we have supplemented GOS into milk replacers to provide better/adequate nutrition for poorly performing piglets, removed from sows, GOS being an essential natural, nutritional component of colostrum/milk during suckling in piglets remaining with the sow. We would welcome the reviewers comments on this.

Round 2

Reviewer 1 Report

The authours have addressed my questions. This manuscript can be accepted in present form.